# Ultrastructural insights into the microsporidian infection apparatus reveal the kinetics and morphological transitions of polar tube and cargo during host cell invasion

Himanshu Sharma[1,2]*, Nathan Jespersen[1], Kai Ehrenbolger[1,2], Lars-Anders Carlson[2], Jonas Barandun [1]*

1 Department of Molecular Biology, The Laboratory for Molecular Infection Medicine Sweden (MIMS), Umeå Centre for Microbial Research (UCMR), Science for Life Laboratory, Umeå University, Umeå, Sweden,
2 Department of Medical Biochemistry and Biophysics, The Laboratory for Molecular Infection Medicine Sweden (MIMS), Wallenberg Centre for Molecular Medicine, Umeå Centre for Microbial Research (UCMR), Umeå University, Umeå, Sweden

* himanshu.sharma@umu.se (HS); jonas.barandun@umu.se (JB)

**Data Availability Statement:** The sub-tomogram averages can be accessed at the Electron Microscopy Data Bank with the following accession

## Abstract

During host cell invasion, microsporidian spores translocate their entire cytoplasmic content through a thin, hollow superstructure known as the polar tube. To achieve this, the polar tube transitions from a compact spring-like state inside the environmental spore to a long needle-like tube capable of long-range sporoplasm delivery. The unique mechanical properties of the building blocks of the polar tube allow for an explosive transition from compact to extended state and support the rapid cargo translocation process. The molecular and structural factors enabling this ultrafast process and the structural changes during cargo delivery are unknown. Here, we employ light microscopy and in situ cryo-electron tomography to visualize multiple ultrastructural states of the *Vairimorpha necatrix* polar tube, allowing us to evaluate the kinetics of its germination and characterize the underlying morphological transitions. We describe a cargo-filled state with a unique ordered arrangement of microsporidian ribosomes, which cluster along the thin tube wall, and an empty post-translocation state with a reduced diameter but a thicker wall. Together with a proteomic analysis of endogenously affinity-purified polar tubes, our work provides comprehensive data on the infection apparatus of microsporidia and uncovers new aspects of ribosome regulation and transport.

## Introduction

Microsporidia are a group of obligate intracellular parasites that infect hosts across the animal kingdom [1–4]. To propagate from host to host, these fungi-like pathogens form stable environmental spores of prokaryotic size [5]. In addition to their small size, these parasites exemplify reductive evolution in eukaryotes [6–8]. The deletion of many genes considered essential for eukaryotic function has produced some of the smallest known genomes of the kingdom [9], streamlined biochemical pathways [10], and minimized cellular macromolecular

codes EMD-17391, EMD-17468 and EMD-17467. Representative tomograms and the raw tilt series have been uploaded to the Electron Microscopy Public Image Archive (EMPIAR) and can be accessed with the deposition EMPIAR-11557. Mass-spectrometry data has been uploaded to PRIDE under project accession PXD042571.

**Funding:** This work was supported by the H2020 MSCA fellowship (101033469 to HS), Integrated Structural Biology fellowship (JCK-1918 to NJ), H2020 European Research Council (948655 to JB) and the Swedish Research Council (2019-02011 to JB). The funders had no role in study design, data collection and analysis, decision to publish, or preparation of the manuscript.

**Competing interests:** The authors declare no competing interests.

**Abbreviations:** cryo-ET, cryo-electron tomography; EM, electron microscopy; FASP, filter-aided sample preparation; HCD, higher energy collisional dissociation; MMTS, methyl methanethiosulfonate; PTP, polar tube protein; RBL, ricin-B lectin; RP, ribosomal protein; SDC, sodium deoxycholate; SP, signal peptide.

complexes. Recent structural insights into microsporidian ribosomes and proteasomes bound to dormancy factors in the extracellular spore stage have highlighted the importance of mechanisms to enter and exit dormancy [11–14]. This trend of reductive evolution might have emerged from an obligate intracellular lifestyle, while the atypical regulatory factors support pathogen survival in nutrient-limiting environments [15].

While a general trend for reductive evolution is pervasive, microsporidia have also evolved specialized mechanisms for invading and hijacking host cell systems [16,17]. These mechanisms include, for example, an expanded repertoire of nucleotide transporters to steal energy and metabolic precursors from host cells [18,19]. However, the most drastic invention and specialization is the polar tube, a microsporidia-specific organelle used for host invasion. The polar tube comprises at least 6 polar tube proteins (PTPs) that may localize to its outermost layer or the terminal tip [20–22], but the structure and organization of PTPs and the exact composition of the tube is still a mystery. Inside the spore, this organelle adopts a spring-like arrangement and is referred to as the polar filament [23–25]. Polar filament coils can range from a couple to dozens, depending on the organism [16]. Upon exposure to the right environmental stimuli, the spores germinate. This includes an explosive firing at the apical pole of the spore that transforms the polar filament into the extended, tube-like state referred to as the polar tube.

During germination, the entire infectious cellular content, known as the sporoplasm, is pushed through the narrow polar tube and delivered into or close to the host cell [26]. These events, including tube eversion and passage of sporoplasm cargo through this constricted tube, are extremely fast and occur within 2 s [25]. To achieve successful firing, sporoplasm delivery, and host infection, the polar tube and the cargo undergo drastic remodeling during this discharge [25,27,28]. These swift ultrastructure changes in the polar tube evince its extraordinary mechanical properties likely conferred by its components, the PTPs. However, the mechanisms driving polar tube firing, sporoplasm discharge, and the dynamics of polar tube eversion are not well understood [25,28–30]. Further, the swift nature of germination events creates challenges in capturing polar tube dynamics and contributes to the generally understudied nature of these parasites. Additionally, most in vitro studies on the microsporidian infection apparatus using electron microscopy (EM) employ denaturing purification of polar tube components or have relied on non-native staining and resin embedding methods, thus limiting their overall resolution.

To overcome these challenges, we employed light microscopy and cryo-electron tomography (cryo-ET) to examine the dynamics and ultrastructure remodeling of the *Vairimorpha necatrix* polar tube during germination in a near-native environment. We capture snapshots of the infectious sporoplasm seen as arrays of organized ribosomes and densely packed cargo passing through the polar tube. Further, 2 distinct states of the extruded polar tube, discernible in the protein layer lining its outer wall, are also described. We further characterize the outer proteinaceous layer of the tube using affinity purification and mass spectrometry, thus unraveling potential new protein–protein interactions of the PTPs. Overall, our results shed new light on the events underpinning cargo delivery by the polar tube and uncover protein factors that may assist host invasion.

## Results

### Cryo-ET captures assorted states of the polar tube and cellular content during cargo delivery

The microsporidian infection apparatus rapidly transitions from a tightly packaged polar filament to an extended polar tube. The dynamics of germination and polar tube length have

been studied in 3 human-infecting microsporidia; however, spore firing is known to vary between species, and firing efficiencies can also vary greatly even for uniform-looking spores [25]. We set out to identify and optimize conditions for *V. necatrix* germination and quantify the dynamics of tube firing in this agriculturally important parasite of Lepidoptera [31,32]. We used live microscopy to record 53 germination events, which enabled us to measure the polar tube length and firing velocity (**S1 Fig**). We observed that polar tubes could attain a maximum length of approximately 142 μm, a mean length of 113 μm, and extend with a mean maximum velocity of approximately 281 μm/s. Similar to the previous observations in distant microsporidia [25], *V. necatrix* everts its tube to the maximum length in less than 1 s and expels the sporoplasm from the tip of the nascent polar tube. Additionally, tube firing follows a typically observed triphasic mode where the tube undergoes observable states of elongation, followed by stasis during sporoplasm passage and a refractory period where the tube shortens after cargo emergence [25] (**S1 Fig**). Akin to tube remodeling events, the sporoplasm also likely transforms from a restricted spore state to extended conformation during extrusion [30], into a spherical shape upon emergence (**S1a Fig**), and these events may impose immediate reorganization of subcellular structures.

The optimal alkaline priming conditions (see Methods) for polar tube firing were used to cryogenically preserve on-grid germinated spores (**S2a Fig**) for in situ characterization of polar tubes and cargo using cryo-ET. On-grid germination was optimized with spores consistently displaying high germination efficiencies (>80%). Spores were applied onto cryo-grids immediately after resuspension in the germination buffer, followed by blotting and plunge freezing. Various grid types were screened during freezing attempts, and polar tubes frequently interacted strongly with the regular grid support regions while avoiding thin carbon films or holes. To overcome this, we utilized lacey carbon grids with an ultra-thin carbon film where the variability in hole size and support mesh increased the chance of trapping sections of the tubes on larger thin-carbon areas. Subsequently, we collected 50 tilt series from different sections of polar tubes (**S1 Table**), of which 45 were suitable for tomogram reconstruction using the IMOD package [33]. The reconstructed tomograms were denoised using IsoNet [34] and auto-segmented using CNN in Eman2 [35] to better visualize the spatial organization of the cargo and polar tubes. This allowed us to capture a wide range of polar tube ultrastructural states, with differences in diameter, outer layer thickness, and interior composition, most likely resulting from the differential passage of reorganized sporoplasm.

The diameter of the analyzed polar tubes ranges from 60 nm to 190 nm depending on their internal content, suggesting we captured a range of different phases during germination or after sporoplasm translocation (**Fig 1**). Polar tubes filled with dense cellular cargo (*PTcargo*) (**Fig 1A–1C**) had diameters of more than 120 nm and are enveloped by a thin layer; in contrast, the electron-lucent or empty tubes (*PTempty*) (**Fig 1D and 1E**) have a diameter of significantly less than 120 nm but were enveloped by a thick outer layer (**S2b Fig**). Here, it is worthwhile to note that the *PTcargo* or *PTempty* states represent static snapshots after firing and could represent states before or after sporoplasm translocation. Among the 2 states, of note was an apparent inverse correlation between the PT diameter and the thickness of the tube wall, which is composed of a lipid bilayer (pink arrows, **Figs 1** and **S2**), flanked by an outer protein layer (light blue and magenta arrows, **Figs 1** and **S2**). These observations are also in line with previous reports of an outer proteinaceous layer in germinated PTs [29,36,37]. Membrane-less PTs were not observed, and the PT membrane appeared continuous in all raw tomograms (small discontinuities in the membrane auto-segmentation likely being the result of mis-annotation due to limited signal-to-noise ratio). Most interestingly, across all our tomograms, we observed different internal cargo, including electron-dense material inside membranous compartments, randomly oriented or highly ordered large molecular complexes such as

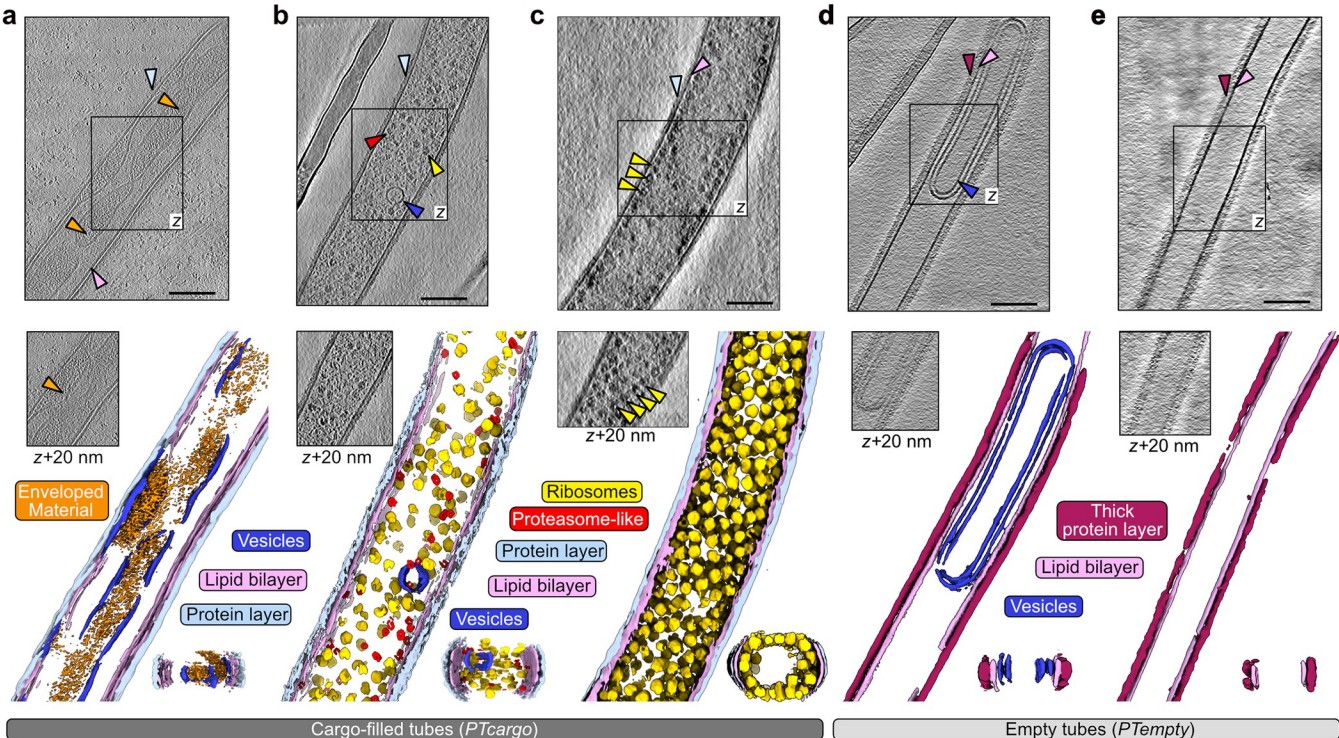

**Fig 1. Ultrastructural states of the polar tubes during or after delivering cellular cargo. (a–e)** Slices through cryo-tomograms of germinated polar tubes above the respective neural network aided 3D segmentation. The central slice of a tomogram is represented at the top in each panel, as seen from the z-axis view. An additional section from the same tomogram is marked by a boxed region and shown below at axis z + 20 nm. The neural network-aided 3D segmentations corresponding to each tomogram are presented in the lower panel. Selected regions of interest in the tomogram slices, such as cellular complexes (a–c), the outer protein layer (a–e), or empty vesicles (d), are indicated with arrows colored as the labels and segmentations below. The black scale bars correspond to 100 nm.

ribosomes and proteasome-like particles, and empty vesicles or empty tubes. The electron-dense cellular material seemed free-flowing or sometimes enveloped inside vesicle-like organelles (orange arrows, **Figs 1A and S2d**), while distinct densities corresponding to macromolecular complexes were randomly distributed or arranged in a very regular fashion (**Fig 1B and 1C**). Vesicle-containing tubes were reminiscent of the frequently observed tube-inside-tube architecture of the polar tube [29]. Notably, macromolecular complexes were completely missing from *PTempty*, which mostly housed electron-lucent vesicles (dark blue arrows, **Figs 1D and S2c**) or no vesicles (**Figs 1E and S2c**). Collectively, these tomograms capture the myriad heterogeneous states the polar tubes and the cellular cargo adopt immediately upon firing.

## Regular clustering of ribosomes in the cargo-filled polar tubes

In 5 out of the 45 cryo-ET reconstructions, we captured sporoplasm sections with an unusually high concentration of ribosome-like particles (**Figs 2, S2E, and S2F and S1 Video**). The particles were arranged in an array-like pattern (**Fig 2A**), wherein their parallel alignment formed a helix traversing the entire length of the reconstruction (**Fig 2B**). These arrays were prominent in the distal sections of the tomograms and the y-axis view of the tube cross-sections, indicating the spirals line the inner wall of the tube (**Fig 2A and 2B**). To confirm their identity, we performed reference-free subtomogram averaging on the spirally arranged ribosome-like particles and obtained a low-resolution map with clear ribosome features (**Figs 2C and S3A**). This map superimposes well with the published high-resolution structure of the *V. necatrix*

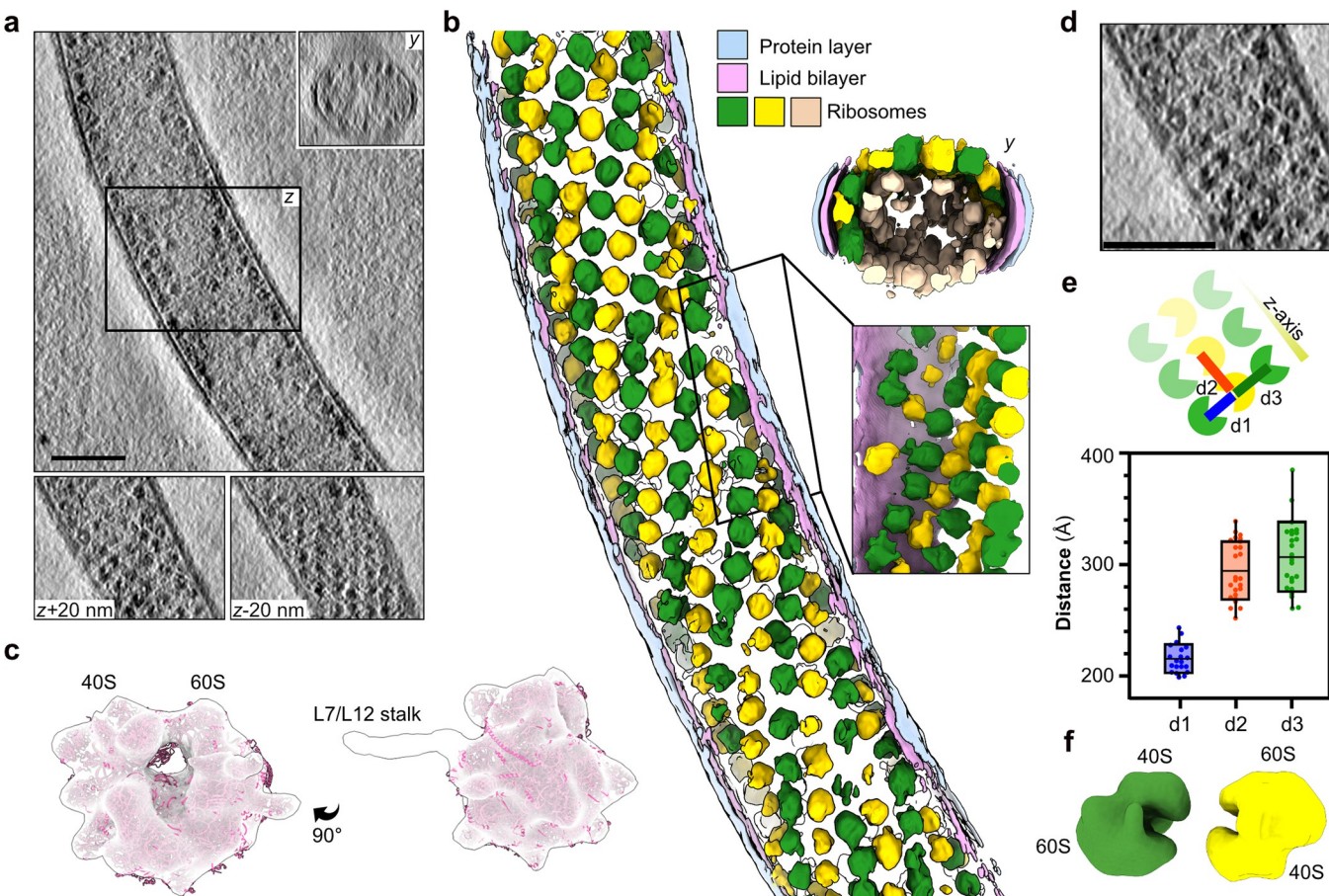

**Fig 2. Spiral-like arrays of ribosomes clustering in proximity to the tube wall.** (**a**) Slices through a cryo-tomogram and the corresponding neural network aided 3D segmentation (**b**) of a polar tube filled with clustered ribosome arrays. The segmentation presents ribosomes in yellow, green, or beige. The yellow and green colored particles indicate the front view from the xy-plane, while the beige-colored particles cluster at the distal end of the same plane. The lipid bilayer is shown in pink, while the outermost protein layer is shown in blue. A y-axis view of the tomogram, as well as slices at z+20 nm and z-20 nm, are shown in the inset. A similar view along the y-axis is depicted with the segmented tube. The scale bar is 100 nm. (**c**) The subtomogram average of clustered particles picked from cargo-filled polar tubes. The map is lowpass filtered to at a resolution of 50 Å and is superimposed with the structure of the *V. necatrix* ribosome (PDB ID: 6RM3, magenta). (**d**) A zoomed-in view of the ribosome array packing inside the polar tube. The scale bar is 100 nm. (**e**) A schematic representation of panel (d) with the inter-ribosome distances (d1–d3) indicated (top) and plotted (bottom). The scatter plot depicts the average distance and distribution between particle centers measured from 3 tomograms containing ribosome spirals. Each dot represents a ribosome pair chosen for measuring the distances. The distance d1 corresponds to 2 close particles, while d2 and d3 are the more distant. The raw data underlying this figure can be found in S1 Data. (**f**) A subtomogram average of ribosome dimer particles is shown lowpass filtered to a resolution of 50 Å and colored as in (b).

ribosome [13] (**Figs 2C and S3B**). We could unambiguously identify features of the 40S and 60S ribosomal subunits thus affirming that the spirally arranged particles are indeed ribosomes (**Fig 2C**). Among these features, density likely corresponding to parts of the well-conserved L7/L12 stalk region was also observed. To further understand the overall organization of the ribosome spirals, we measured the inter-particle distance between the 4 nearest neighboring macromolecules (**Fig 2D and 2E**). The nearest particle pairs approach 200 Å, with a mean interparticle distance of 220 Å (d1), spread uniformly across the tube. In contrast, the other adjacent ribosomes (d2, d3) are approximately 300 Å distant, on average. An inter-ribosome distance of 220 Å matches well with the observed 100S hibernating ribosome dimer from *Spraguea lophii* [38], suggesting that the parallel spiral is composed of repetitive sets of hibernating ribosomes organized into an almost crystalline-like arrangement inside the tube. To further confirm this observation, we performed subtomogram averaging on particles picked from the

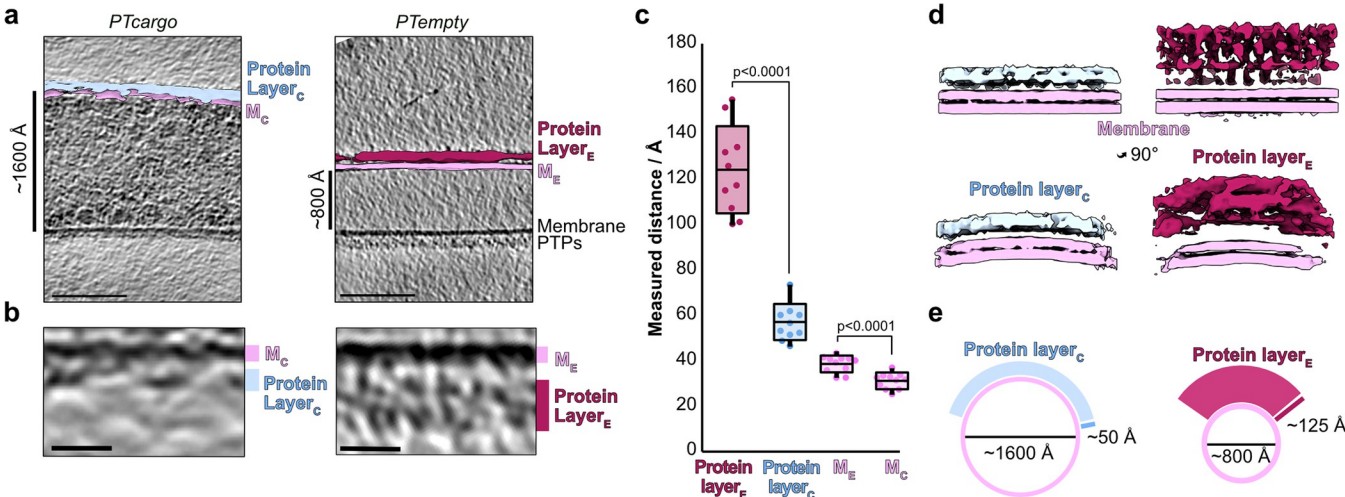

**Fig 3. Visualizing polar tube wall features from cargo-filled and empty tubes.** (**a**) Central slices through representative cryo-tomograms from cargo-filled and empty polar tubes. Segmentations of the inner and outer layers are superimposed onto 1 side of the tube in both tomogram slices. The scale bar is 100 nm. (**b**) Zoomed in sections of the tube wall from (a). The scale bar is 10 nm. (**c**) A plot depicting the thickness of each outer protein layer and membrane bilayer, as measured across various tomograms, for cargo-filled and empty polar tubes. The $p$-values of two-tailed, unpaired Student's $t$ test analyses are shown above the compared plots. The raw data underlying this figure can be found in S1 Data. (**d**) Side-by-side comparison of 2 slab views of subtomogram averages obtained from cargo-filled (left) or empty (right) polar tube walls. The volumes were segmented and colored by membrane and outer layer. The views at the top (slabs along the polar tube) are shown below rotated 90° around the Y-axis (slabs across the polar tube). Map regions are colored as in (a). (**e**) A schematic representation of polar tube and outer protein layer remodeling during cargo movement.

spiral, which converged to a low-resolution map representing ribosome dimers (**Figs 2F, S3D, and S3E**), as observed in *Spraguea lophii* [38].

## Adaptation in the polar tube outer wall during cargo translocation

Inspecting tomograms of PT segments in cargo-filled and electron-lucent states allowed us to compare the ultrastructural organization of the outer layer in empty and filled tubes (**Figs 1A–1E and 3A**). Since the presence or absence of cargo significantly affected the tube diameter, we assumed this could also affect the structure of the outer layer. To investigate this, we compared the coats of cargo-filled and empty tubes by manually measuring the thickness of the individual layers and by subtomogram averaging of the tube wall from both states. In our tomograms, we could resolve 2 individual layers where the inner layer had 2 electron-dense leaflets suggesting it is a lipid bilayer membrane. In contrast, the outer layer was less electron-dense, and we hypothesize that this layer is composed of proteins, some of which are PTPs, in line with previous studies [36]. The presence of an outer protein-composed tube wall also agrees well with immunolabeling studies where fluorescently labeled antibodies against individual PTPs localized to the outer sheath of the tube wall [20]. We surveyed the thickness of the inner lipid bilayer membrane and the outer protein-composed layer (**Fig 3B**) across several tomograms and compared the measured values between empty and cargo-filled tubes. The thickness of the lipid bilayer was, approximately 30 to 40 Å, relatively consistent between both states (**Fig 3C**). In strong contrast, the protein layer thickness varied significantly between the 2 polar tube states where the outer layer was around 50 Å thick in all measured regions of *PTcargo* but ranged from 100 to 150 Å in *PTempty* (**Fig 3C**).

Further, we performed subtomogram averaging with sections extracted from the outer layer of the empty and cargo-filled tubes. The averages derived from the 2 ultrastructural states of the tube resolved the membrane into 2 density layers as expected for a lipid bilayer (**Figs 3**

and S3C), in good agreement with the known thickness of membranes [39]. In addition, the subtomogram averages reiterate the differences observed in protein layer thickness (Fig 3D). This suggests the protein wall has unique mechanical properties that allow it to stretch out into a thin layer when cargo traverses through the tube and contract again once the sporoplasm is ejected at the tip (Fig 3E). These observations collectively suggest the importance of a dynamic PTP containing protein layer that undergoes major structural remodeling to facilitate polymorphic states of the polar tube during cargo delivery (Fig 3E).

## Native composition of the outer protein layer

Intrigued by the large-scale remodeling of the polar tube, we focused on understanding its composition. For this, we discovered a stretch of 8 proximate histidines, serendipitously located in the known polar tube component PTP3 from *V. necatrix* (Fig 4A). Generally, PTP3 is the largest among known polar tube components and is predicted to harbor a cleavable signal peptide and stretches of disordered regions. PTP3 antibodies localize to the entire length of the polar tube except for the tip [40]. This suggests that PTP3 is part of the protein scaffold of the tube, which makes it an ideal candidate for endogenous compositional analysis. We affinity-purified endogenous PTP3 and its potential interaction partners for protein identification analysis using mass spectrometry (Fig 4B and S2 Table). In this sample, PTP3 was significantly enriched and the most abundant protein (Fig 4C–4E and S3 Table), confirming the

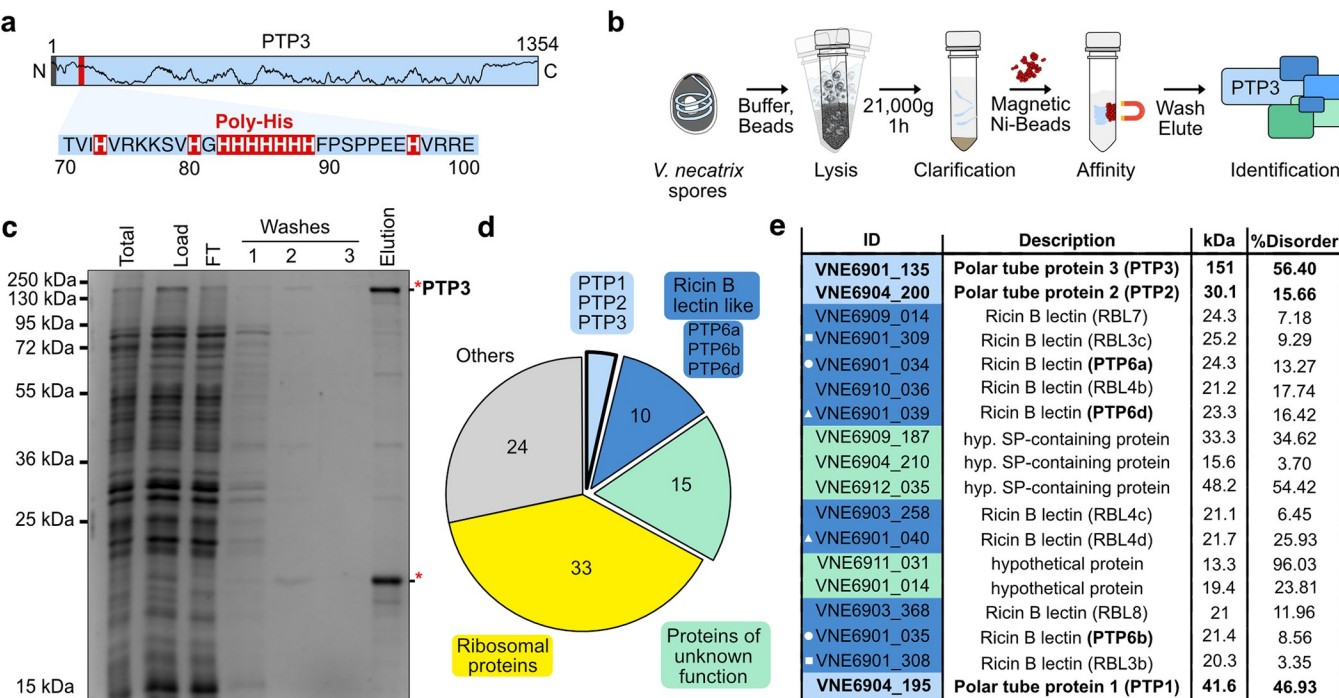

**Fig 4. Endogenous PTP3 pulldown.** (**a**) Representation of the 1,354 amino acids long *V. necatrix* PTP3. Marked are the N-terminal predicted signal peptide (highlighted in gray), the histidine stretch (red area zoomed in with sequence) that was used for purification, and the disorder prediction (black line). (**b**) Schematic representation of the endogenous purification of PTP3. (**c**) The different purification steps shown in (b) have been analyzed on an SDS PAGE. See the S1 Raw Image file for the uncropped gel. The 2 bands in the "Elution" lane (indicated by the red stars) were excised for in-gel mass spectrometric analysis. (**d**) A pie chart showing the distribution of all mass spec hits detected in the elution sample, shown in (c), classified using functional and structural annotation of the *V. necatrix* genome (manuscript in preparation). (**e**) Mass spec hits from the classes: PTPs, RBLs, and proteins of unknown function, are sorted based on the number of significantly detected unique peptides (high to low). Gene IDs are presented with colors for the respective classes, along with names for the encoded proteins, molecular weights, and structurally disordered regions predicted using PrDos [75]. The white symbol in front of the gene ID indicates close genomic localization. The full mass spectrometry data table can be found in S2 and S3 Tables.

utility of the endogenous poly-histidine patch for affinity purification. The known polar tube members PTP1 and PTP2 were also captured and present in our pull-down, where PTP2 was the second most enriched protein among all hits (**Fig 4E**). This suggests direct or indirect interaction of PTP1 and PTP2 with PTP3, which agrees with previous yeast two-hybrid assays for PTPs from *Encephalitozoon cuniculi* [41]. However, neither PTP5 nor the polar tube tip localizing PTP4 [20] could be detected, suggesting the absence of direct or stable interactions with PTP3. This is also consistent with recent observations that PTP3 is not present at the tip of the tube [37].

In addition to the known polar tube members, other enriched proteins in our mass spectrometry analysis suggest additional factors may constitute the microsporidian infection organelle. The remaining hits were segregated based on direct functional assignment or the presence of signature structural features (**Fig 4D**). A significant class with ricin-B lectin domain-containing proteins (RBLs) accounts for more than 10% of all hits. After PTP3, several RBLs were the next most enriched proteins in the pulldown and may correspond to proteins migrating around 20 kDa in the gel (**Fig 4C and S3 Table**). RBLs contain the B-chain of the ricin glycoprotein and enable interactions with β-linked galactose or N-acetylgalactosamine containing glycans [42] and assist infection by promoting host cell adherence [43,44]. We noted that the tandem genomic location for some of the detected RBL encoding genes agrees well with the previously observed synteny in microsporidian genomes [45] and may indicate the presence of dimers (e.g., VNE6901_034/035, VNE6901_039/040, VNE6901_308/309). Interestingly, among these RBLs, VNE6901_034, VNE6901_035, and VNE6901_039 possess sequence and structure homology to *Nosema bombycis* PTP6 (**S4 Fig**), also known to localize throughout the polar tube [22], suggesting the presence of PTP6 isoforms and their interaction with PTP3.

Additionally, several proteins of unknown function (>25%), containing spore wall-like proteins, proteins with signal peptides (SPs), or high propensity of disordered regions, were identified (**Fig 4E and S2 Table**). Finally, the remaining hits could be attributed to ribosomal proteins (RPs) and other proteins involved in cellular maintenance (**S2 Table**). Although our work discovered ribosomes that cluster along the inner PT wall (**Fig 2**), we assume that the RP hits represent contaminations, as their high abundance and the unusual biochemical properties often lead to the identification of peptides from RPs via mass spectrometry. Collectively, this mass spectrometry data, to our knowledge, provides the first endogenous compositional analysis of an affinity-purified polar tube and suggests novel interactions between RBLs and PTPs, and also uncovers new candidates that may assist polar tube structure and function.

## Discussion

Host cell invasion is an understudied aspect of the microsporidian lifecycle that initiates with polar tube firing to inoculate host cells with the infective sporoplasm [26]. The mechanisms enabling tube remodeling, ensuing sporoplasm translocation through an extremely constricted passage, and the state of sporoplasm during delivery remain unaddressed. Here, we probe cargo delivery by cryogenically preserving snapshots of the ultrarapid spore germination process. Congruence in polar tube firing kinetics seen in our work and previous studies [23,25], suggest a conserved theme of sporoplasm movement upon germination (**S1 Fig**) between different microsporidian species. Firing does not occur simultaneously across all spores, and the implementation of on-grid germination successfully captured heterogeneous states of the PT and sporoplasm by immediately vitrifying spores upon inducing firing. It is possible that the mechanical interaction of the PT with the grid support limited tube flexibility, thus slowing down the rapid firing, and trapped otherwise transient states. Two caveats are worth mentioning in relation to this analysis. Firstly, any ordering of the observed structural states onto a

timeline is only hypothetical, since cryo-ET does not provide any time resolution. Secondly, we cannot completely rule out that some tomograms represent misfired tubes that fail to fully deliver cargo [25]. However, we made deliberate efforts to avoid imaging such PTs by actively excluding tubes that exhibited bloated appearances and extremely high electron density during the data collection process.

The PT regions seen in our tomograms show the heterogeneous states of free-flowing macromolecular complexes, vesicles of varying electron density (**Fig 1**), and spirally packed hibernating ribosomes (**Fig 2**), traversing the tubes. The presence of membranous vesicles and similarly encased electron-dense material suggests the maintenance of compartmentalization even during explosive events that mandate rapid cellular deformations to facilitate cargo movement in the tube. Another type of non-membranous compartmentalization is seen as long strands of highly organized ribosome spirals in some fired tubes (**Fig 2**). Here, the occurrence of free flowing (**Fig 1B**), as well as spirally packed ribosomes (**Figs 1C and 2**) in germinated tubes likely represents 2 dynamic states that could interconvert during host invasion. Similar crystalline arrays of eukaryotic ribosomes, formed as part of a stress response [46–48] or upon chemical treatments [49], display a reversible-metabolically inactive state and resistance to nucleolytic degradation [50]. Collectively, akin to hibernating ribosome dimers in bacteria [51,52], the spiral of ribosome dimers in microsporidia may aid in ribosome hibernation, further adding to the atypical translation regulation mechanisms seen in microsporidian ribosomes [11,13]. Alternatively, these spirals could represent a transient state of polyribosomes seen in pre-spore stages of intracellularly growing microsporidia [53] and precursors to the free hibernating ribosome dimers [38]. This would also provide an efficient ribosome packing, transportation, and redeployment strategy, wherein rapidly translocating arrayed ribosomes may induce a local ribosome crowding [54–56] for a translation jumpstart to maximize protein synthesis rates. Similarly, compartmentalization would facilitate the immediate restructuring of sporoplasms after host cell invasion to expedite the hijacking of host functions.

Our work confirms previous observations [57] that the polar tube undergoes large-scale structural reorganization during germination. Most dramatically, the tubes seem to nearly double in internal diameter and quadrupled tube volume in the *PTcargo* state (**Fig 3**). Collectively, these observations suggest that both tube ultrastructure and sporoplasm reorganizations [25] are contemporaneous events facilitating sporoplasm delivery. Here, the remodeling of a highly dynamic tube protein layer around the fluid lipid bilayer drives these large-scale ultrastructure changes (**Fig 3**), likely in response to the hydrodynamic pressure exerted by the traversing sporoplasm. Such instances of mechanical recoil are widely reported in elastins and other membrane-localizing disordered proteins involved in mechanical and trafficking functions in cells. Interestingly, PTP1 and PTP3 and other factors identified in our native pulldown experiments (**Fig 4**) contain a significant fraction of predicted disorder. The high structural flexibility of disordered proteins provides a large conformational sample space for structure and membrane curvature remodeling under tension [58–60]. Further, the observed posttranslational O-glycosylation and mannosylation of PTPs [36,41,61] may further expand their protein–protein interaction repertoire, structural flexibility [62], and host cell attachment abilities [63–65]. Additionally, our native purifications suggest an association of PTPs with several carbohydrate-binding RBLs that promote host-cell attachment in microsporidia and other intracellular pathogens [43,66–68]. Further, the presence of signal peptides-containing hypothetical proteins suggests extracellular localization and potential PTP interaction to form the outer, flexible layer of the polar tube.

In summary, our study provides structural insights into the germinated microsporidian polar tube, revealing a spiral array-like arrangement of ribosomes, 2 distinct structural states

of the protein coat of the polar tube, and new potential components of this layer. Further validation and biochemical characterization of these potential PTPs and a higher resolution structure of the polar tube components will help us fully understand the mechanistic basis of tube reorganization and the role of individual PTPs and RBLs in host cell invasion.

## Methods

### Spore isolation

*V. necatrix* was cultivated and reproduced by feeding approximately 100,000 spores to fourth and fifth instar *Helicoverpa zea* larvae grown on a defined diet (Benzon Research). After 3 weeks at 21 to 25˚C, the spores were harvested. First, larvae were homogenized in water, followed by filtration through 2 layers of cheesecloth and subsequent filtration through a 50 μm nylon mesh. The filtrate was layered on top of a 50% Percoll cushion in a 2-ml microcentrifuge tube, and spores were pelleted by centrifugation at 1,000g for 10 min. The pure spores were stored at −80˚C until further use.

### Germination of *V. necatrix* spores

In our study, spore preparations for *V. necatrix* displayed some batch-to-batch variation in germination efficiencies. Hence, a thorough analysis of germination efficiency using alkaline priming [32] was tested out to select a batch for cryoET work. Briefly, spores were incubated in 100 μl of 0.01 M KOH for 15 min at room temperature, followed by pelleting via centrifugation at 10,000g for 2 min. Primed spores were then germinated by resuspending in 100 μl germination buffer (0.17 M KCl, 1 mM Tris-HCl (pH 8.0), 10 mM EDTA). Germination was confirmed by light microscopy. In the batch used for cryo-ET, approximately 80% of spores germinated with the immediate addition of the germination buffer. Additionally, in 9% of the monitored germination events, sporoplasms were not ejected from PTs and tube lengths did not decrease after reaching their maximal length. As the goal of this study was to describe and characterize successful germination conditions, these incomplete germination events were excluded from all analyses.

### Light microscopy

To examine the germination process via light microscopy, 0.1 mg of alkaline-primed *V. necatrix* spores were resuspended in 50 μl of the germination buffer. Next, 2.5 μl of spore suspension was immediately transferred to a glass slide and sealed with a cover slip. Germination occurred up to 5 min after resuspension in the germination buffer. Videos of polar tube firing were captured using a Nikon 90i Eclipse microscope equipped with a 40× PH2 phase-contrast objective lens and a Hamamatsu C4742-80 ORCA-ER digital camera, collecting images at 15 frames per second.

Kymographs were produced using the straightening function within the FIJI image analysis software [69]. To quantify the lengths and maximal velocities, polar tube length was measured on a frame-by-frame basis in the FIJI software using the segmented line function, starting with the polar tube exit site on the germinating spore. Maximal velocity was calculated as the point of greatest change in length per frame.

### On-grid germination of spores and data collection

Alkaline-primed spores were resuspended in the germination buffer and immediately blotted onto glow-discharged Lacey Carbon 200-UT grids. Grids were blotted with a blot time of 3.5 to 4.5 s with a blot force of −1 using an FEI Vitrobot Mark IV (Thermo Fisher Scientific),

followed by plunge freezing in liquid ethane. The vitrobot was set to 4˚C and 100% humidity throughout the blotting process. Tilt series were collected on a Titan Krios (Thermo Fisher Scientific) operated at 300 kV using a Gatan K2 BioQuantum direct electron detector at the Umeå Centre for Electron Microscopy. Tilt series were collected on the Tomo5 package (Thermo Fisher Scientific) using an object pixel size of 2.173 Å. A variable tilt angle range of −60˚ to +60˚ with a step range of 2˚ or 3˚ was used for a dose-symmetric data collection scheme [70]. A nominal defocus range of −1.5 to 5 μm was utilized across the data collections, the total electron dosage was fixed at 110 to 120 e-/Å$^2$, and a total of 50 tilt series were collected, of which 45 were used for analysis. Positions for tilt series collection were chosen by visual inspection of tube regions based on a heterogenous mix of thickness and electron density of visible features. Spores and polar tube tips with ejected cargo were too electron dense for collecting tilt series and hence were excluded from data collection.

## Tilt series processing, tomogram generation, segmentation, and distance measurements

Tilt series motion correction was done using MotionCor2 [71]. Tilt series alignment was done using patch-tracking in IMOD followed by CTF correction using CTFPlotter [33]. Dose filtering and reconstruction were performed using the IMOD package. Tomograms were reconstructed using the weighted back projection and subsequently binned 4 times for analysis. Tomograms were denoised via Isonet filtering [34] and imported into EMAN2 for CNN-based picking and segmentation [35]. EMAN2's inbuilt, default 4-layer CNN architecture was utilized for training networks for annotating features on the tomogram. For segmentation, default pipelines for importing and preprocessing tomograms were used. Areas of interest were picked with a box size of 64, and a pen size of 1 was used for annotating the particles. The ratio of the number of particles of interest to negative reference was roughly 1:20, and all network training parameters were kept unchanged. All particles of interest were segmented using separate CNN-based models, and the output from each model was grouped using Segger [72], and visualized in ChimeraX. The binned, denoised tomogram projections were also used to manually measure membrane thickness and inter-ribosome distances using IMOD and Dynamo. For measurements on PT outer layers, 10 tomograms each from *PTempty* and *PTcargo* were surveyed uniformly across their length to calculate the mean thickness for each tomogram (*n* = 10, for each tomogram) (**Figs 3C** and S**5**).

## Subtomogram averaging

Subtomogram averaging was carried out as schematically indicated in (**S3 Fig**). Around 150 particles were manually picked and extracted from two 4-times-binned tomograms using Dynamo [73]. The resulting particles were manually centered and aligned by visual inspection in the dgallery function in Dynamo. Manually aligned particles were then averaged to generate an unaligned average. Subsequently, the inbuilt template-matching function in Dynamo was used to pick particles using a low-pass filtered initial model. Post template matching, picks were selected from individual tomograms based on cross-correlation scores upon manual inspection. The final coordinates tables were cleaned to retain picks arising from the spiral arrangement, and approximately 3,000 particles were extracted for the final alignment. A spherical mask centered on the initial model was created for alignments, and alignment was performed with limiting shifts and allowing for full-azimuthal and full-in-plane rotations. Azimuthal angles of the particles in the crop table were then randomized to decrease the impact of the missing wedge, and by this process, another average was generated. Full-azimuthal rotations and limited (±60˚) in-plane rotation were performed with C1 symmetry for the final

averaging round. The resolutions were estimated to 49 Å using the Gold-standard Fourier shell correlation with a threshold of 0.143. A similar methodology was utilized for subtomogram averaging of ribosome dimers picked along the inter-ribosome distance d1. However, averaging attempts for ribosome dimer repeatedly generated anisotropic subvolumes, likely due to a specific orientation problem.

For subtomogram averages of sections of the polar tube wall, Dynamo's inherent surface models were generated manually for each tomogram and utilized for defining initial orientation before particle extraction. Extracted particles from 1 tomogram of each *PTcargo* and *PTempty*, respectively, were averaged without alignment to create initial models. Subsequently, particles were extracted from all tomograms. A first round of alignment run was performed using the initial model as a template with angular sampling similar to the ribosome alignment described above, along with removing oversampled particles based on the "separation in tomogram" parameter. Particles averaged after the removal of duplicates were extracted to create a new table and used for the final alignment. Resolution estimates using the Gold-standard Fourier shell correlation with a threshold of 0.143 for *PTempty* segments stood at 26 Å but could not be determined reliably for *PTcargo* segments.

## Affinity purification of PTP3 and associated proteins

To enrich a polar tube sample, 50 mg of *V. necatrix* spores were resuspended in 500 μl lysis buffer (50 mM Tris (pH 8.0) at 4˚C, 150 mM NaCl, 40 mM Imidazole, 5 mM DTT, protease inhibitor cocktail consisting of PMSF, E64, Pepstatin) and DNAseI. Spores were lysed via bead beating in tubes containing Lysing Matrix E (MP Bio) in 3 × 1-min intervals in a Fast-Prep 24 (MP Bio) grinder at 5.5 m/s with 5-min break on ice in-between. The lysate was transferred into new tubes and centrifuged for 1 h at 21,000g and 4˚C. The clarified lysate was then incubated with 100 μl His Mag Sepharose Ni beads (Cytivia) rotating at 4˚C to capture and enrich *V. necatrix* PTP3 via the poly-histidine patch on its N-terminal end (**Fig 4A**). The beads were washed 3 times with 1 ml wash buffer (50 mM Tris (pH 8.0) at 4˚C, 150 mM NaCl, 80 mM Imidazole, 5 mM DTT) and eluted in 80 μl elution buffer (50 mM Tris (pH 8.0) at 4˚C, 150 mM NaCl, 250 mM Imidazole, 5 mM DTT). The sample was analyzed on SDS-PAGE and sent for in-solution mass spectrometry analysis.

## Proteomics sample preparation

The sample was reduced with DL-dithiothreitol (DTT, 100 mM) at 60˚C for 30 min and digested with trypsin using a modified filter-aided sample preparation (FASP) method [74]. In short, the reduced sample was transferred to a Microcon-30 kDa centrifugal filter (Merck), then washed repeatedly with 8 M Urea, 50 mM triethylammonium bicarbonate (TEAB) and once with digestion buffer (0.5% sodium deoxycholate (SDC), 50 mM TEAB). The reduced cysteine side chains were alkylated with 10 mM methyl methanethiosulfonate (MMTS) in the digestion buffer for 30 min at room temperature. The sample was repeatedly washed with a digestion buffer and digested with trypsin (0.2 μg, Pierce MS grade Trypsin, Thermo Fisher Scientific) at 37˚C overnight. An additional portion of trypsin (0.2 μg) was added and incubated for another 3 h the next day. The peptides were collected by centrifugation, and SDC was removed by acidification with 10% trifluoroacetic acid. The sample was purified using High Protein and Peptide Recovery Detergent Removal Spin Column (Thermo Fisher Scientific) and Pierce peptide desalting spin columns (Thermo Fisher Scientific) according to the manufacturer's instructions. The purified peptide sample was dried on a vacuum centrifuge and reconstituted in 3% acetonitrile and 0.2% formic acid for the LC-MS/MS analysis.

### NanoLC-MS analysis and database matching

Analysis was performed on an Orbitrap Exploris 480 mass spectrometer interfaced with an Easy-nLC1200 nanoflow liquid chromatography system (Thermo Fisher Scientific). Peptides were trapped on an Acclaim Pepmap 100 C18 trap column (100 μm × 2 cm, particle size 5 μm, Thermo Fisher Scientific) and separated on an in-house packed analytical column (75 μm × 35 cm, particle size 3 μm, Reprosil-Pur C18, Dr. Maisch) from 5% to 45% B over 78 min, followed by an increase to 100% B at a flow of 300 nl/min. Solvent A was 0.2% formic acid, and solvent B was 80% acetonitrile, 0.2% formic acid. MS scans were performed at 120,000 resolution, m/z range 380 to 1,500. MS/MS analysis was performed in a data-dependent manner, with a cycle time of 2 s for the most intense doubly or multiply charged precursor ions. Precursor ions were isolated in the quadrupole with a 0.7 m/z isolation window, with dynamic exclusion set to 10 ppm and a duration of 30 s. Isolated precursor ions were fragmented with higher energy collisional dissociation (HCD) set to 30%, AGC target was set to 200%, and the maximum injection time to 54 ms.

Data analysis was performed using Proteome Discoverer version 2.4 (Thermo Fisher Scientific). The raw data was matched against *V. necatrix* proteome derived from a high-quality genome assembly (manuscript in preparation) using Mascot 2.5.1 (Matrix Science) as a database search engine with a peptide tolerance of 5 ppm and fragment ion tolerance of 30 mmu. Tryptic peptides were accepted with 1 missed cleavage, oxidation on methionine was set as a variable modification, and methylthiolation on cysteine was set as a fixed modification. Fixed Value PSM Validator was used for PSM validation.

## Supporting information

**S1 Data. Raw data points used in the main manuscript figures are organized in individual sheets labeled according to the corresponding figure.**
(XLSX)

**S1 Raw Image. Uncropped SDS-PAGE gel used in Fig 4C.**
(PDF)

**S1 Video. Tomographic volume of a ribosome-filled-germinated polar tube overlaid with the corresponding segmentation.**
(MP4)

**S1 Table. Cryo-ET data collection parameters.**
(PDF)

**S2 Table. Mass spectrometry results of a PTP3 affinity-purified sample.** The list of mass spectrometry hits is arranged according to peptide-spectrum match (PSM). The number of consecutive histidines (3xHis, 4xHis, 7xHis) in the protein is listed. PTP3 is the only protein with 7 consecutive histidines.
(PDF)

**S3 Table. In-gel mass spectrometry analysis of PTP3 affinity-purified sample.** The 2 prominent protein bands indicated by an asterisk (*) in Fig 4C, one around 130 kDa and 20 kDa were sliced and analyzed. The list of mass spectrometry hits is arranged according to peptide-spectrum match (PSM).
(PDF)

**S1 Fig. Tracking polar tube eversion to understand germination dynamics and tube length.**
(**a**) A kymograph obtained via live light microscopy analysis of polar tube firing events from

*Vairimorpha necatrix*. The spore at the bottom of the kymograph is denoted by "S" and the sporoplasm ejected on the distal end is indicated as "SP." (**b**) Length over time diagrams of all analyzed polar tube eversion events. The average length over time is colored in green. (**c**) Bar plots of polar tube maximal length (yellow), length at the end (blue), and maximum velocity distribution (gray). The raw data used to create the plots can be found in the **S1 Data**. (PNG)

**S2 Fig. Representative tomograms depicting empty or sporoplasm-packed germinated polar tubes.** (**a**) A schematic representation of the methodology for on-grid freezing and collecting tomograms of germinated polar tubes. (**b**) A graph representing the internal diameter of polar tubes (*PTempty* and *PTcargo*) visualized using cryo-ET. Each dot represents 1 tube, and the line represents the mean diameter. The raw data underlying this figure can be found in the S1 Data file. (**c**) Representative tomograms of *PTempty* or polar tubes filled with electron-lucent material or completely devoid of cellular cargo. The central section of a tomogram is shown with regions of interest indicated with arrows (magenta for the outer wall, pink for the lipid bilayer, and blue for vesicles). (**d–f**) Representative tomograms from *PTcargo* or polar tubes filled with cellular cargo where (e and f) contained ribosome spirals inside tubes. The central section of a tomogram is presented, and regions of interest are indicated with arrows (light blue for the outer tube wall, pink for the lipid bilayer, and yellow for ribosomes). For (e and f), additional views corresponding to the boxed regions and corresponding segmented tomograms are also presented. (PNG)

**S3 Fig. Subtomogram averaging workflow followed using the Dynamo package.** (**a**) Schematic workflow of the subtomogram averaging procedure to generate the ribosome volume. (**b**) Two 60°-related views of ribosome reconstruction (transparent white), fitted with the structure of the *V. necatrix* ribosome (PDB ID: 6RM3, magenta). (**c**) Schematic workflow of the subtomogram averaging procedure used to create the reconstructions of the segments of polar tube outer layers. The scheme is shown for cargo-filled tubes, and a similar methodology was used for empty tubes. (**d**) Subtomogram averaging workflow used to reconstruct dimeric ribosomes from clustered particles in sporoplasm-filled tubes. (**e**) Low-pass filtered subtomogram averages of ribosome dimers placed back into their original location in germinated polar tubes. Averages are shown in yellow, and the central slice of the tomogram slice is shown in gray. (PNG)

**S4 Fig. Identification of PTP6 and its isoforms in PTP3 pulldowns.** (**a**) Alignment of VNE6901_039 and VNE6901_034 with PTP6 homologs from *Nosema bombycis* (R0MBR8_NOSB1), *Vavraia culicis* (L2GVW3_VAVCU), *Nosema ceranae* (C4V7Y1_NOSC), *Spraguea lophii* (S7XK85_SPRLO), *Encephalitozoon hellem* (I6TKU6_ENCHA), *Encephalitozoon romaleae* (I7AT09_ENCRO), *Encephalitozoon intestinalis* (E0S8R2_ENCIT). Protein sequences were retrieved from Uniprot and aligned using Muscle followed by visualization using Jalview. (**b–d**) Alphafold models, below their corresponding pLDDT scores of the top-ranked predictions, for *N. bombycis* PTP6 (b), VNE6901_039 (c), and VNE6901_034 (d). (**e**) Overlay of predicted PTP6 models from (b–d) shown at 90° rotation. Regions predicted with low confidence have been excluded for clarity. (PNG)

**S5 Fig. Individual data points for PTP layer measurements.** A plot showing the distribution of individual data points measured for the thickness of various features of PT*empty* and PT*cargo* tubes (10 tubes each). Thickness was measured on tomographic projections along the

length of the tubes and the mean value for each measurement is indicated for each tube. The individual data points were utilized to derive measurements shown in **Fig 3**. The raw data underlying this figure can be found in the **S1 Data**.
(PNG)

## Acknowledgments

We thank all members of the Barandun laboratory and the Carlson laboratory for the helpful discussions. Further, we thank Michael Hall and Camilla Holmlund for their help with cryo-EM data collection. The electron microscopy data was collected at the Umeå Centre for Electron Microscopy, a node of the Cryo-EM Swedish National Facility, funded by the Knut and Alice Wallenberg, Family Erling Persson and Kempe Foundations, SciLifeLab, Stockholm University, and Umeå University. The authors also thank the Proteomics Core Facility at Sahlgrenska Academy, University of Gothenburg, for the proteomic analysis.

## Author Contributions

**Conceptualization:** Himanshu Sharma, Jonas Barandun.

**Data curation:** Himanshu Sharma, Nathan Jespersen, Kai Ehrenbolger, Jonas Barandun.

**Formal analysis:** Himanshu Sharma, Nathan Jespersen, Kai Ehrenbolger, Lars-Anders Carlson, Jonas Barandun.

**Funding acquisition:** Jonas Barandun.

**Investigation:** Nathan Jespersen, Kai Ehrenbolger, Jonas Barandun.

**Methodology:** Himanshu Sharma, Nathan Jespersen, Kai Ehrenbolger, Jonas Barandun.

**Project administration:** Jonas Barandun.

**Resources:** Jonas Barandun.

**Supervision:** Jonas Barandun.

**Validation:** Himanshu Sharma, Jonas Barandun.

**Visualization:** Nathan Jespersen, Kai Ehrenbolger, Jonas Barandun.

**Writing – original draft:** Himanshu Sharma, Jonas Barandun.

**Writing – review & editing:** Himanshu Sharma, Nathan Jespersen, Kai Ehrenbolger, Lars-Anders Carlson, Jonas Barandun.

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
