## [Editor Report · Decision Letter 0]

27 Oct 2023

Dear Dr Barandun, 

Thank you for submitting your revised manuscript from Review Commons entitled "Ribosome clustering and surface layer reorganization in the microsporidian host-invasion apparatus" for consideration as a Research Article by PLOS Biology. Please accept my apologies for the delay in getting back to you with feedback as we consulted with an academic editor about your submission. 

Your manuscript has now been evaluated by the PLOS Biology editorial staff, as well as by an academic editor with relevant expertise, and I am writing to let you know that we would like to pursue your manuscript further and send it back out for re-review by the original reviewers at Review Commons.

*IMPORTANT* 

After discussions within the editorial team, we would like to consider your manuscript as a Short Report at the journal (https://journals.plos.org/plosbiology/s/what-we-publish#loc-short-reports). During resubmission (details below), we would be grateful if you could please tick 'Short Report' as the article type in the online submission form. 

Before we can send your manuscript to reviewers, we need you to complete your submission by providing the metadata that is required for full assessment. To this end, please login to Editorial Manager where you will find the paper in the 'Submissions Needing Revisions' folder on your homepage. Please click 'Revise Submission' from the Action Links and complete all additional questions in the submission questionnaire.

Once your full submission is complete, your paper will undergo a series of checks in preparation for peer review. After your manuscript has passed the checks it will be sent out for review. To provide the metadata for your submission, please Login to Editorial Manager (https://www.editorialmanager.com/pbiology) within two working days, i.e. by Oct 29 2023 11:59PM.

Kind regards,

Richard

Richard Hodge, PhD

rhodge@plos.org

PLOS

---

## [Decision Letter · Decision Letter 1]

18 Dec 2023

Dear Dr Barandun,

Thank you for your patience while we considered your revised manuscript "Ribosome clustering and surface layer reorganization in the microsporidian host-invasion apparatus" for publication as a Short Report at PLOS Biology. Please accept my sincere apologies for the long delays that you have experienced during the peer review process. This revised version of your manuscript has been evaluated by the PLOS Biology editors, the Academic Editor and the original reviewers at Review Commons.

Based on the reviews, I am pleased to say that we are likely to accept this manuscript for publication, provided you satisfactorily address the remaining points raised by the reviewers, including the assignments of the proteinaceous layer with polar tube proteins and the CNN architecture. In addition, please also make sure to address the following data and other policy-related requests that I have provided below (A-G):

(A) We would like to suggest the following modification to the title: 

“Ultrastructural insights into the microsporidian infection apparatus reveal the kinetics and morphological transitions of polar tube and cargo during host cell invasion"

(B) You may be aware of the PLOS Data Policy, which requires that all data be made available without restriction: http://journals.plos.org/plosbiology/s/data-availability. For more information, please also see this editorial: http://dx.doi.org/10.1371/journal.pbio.1001797

-Supplementary files (e.g., excel). Please ensure that all data files are uploaded as 'Supporting Information' and are invariably referred to (in the manuscript, figure legends, and the Description field when uploading your files) using the following format verbatim: S1 Data, S2 Data, etc. Multiple panels of a single or even several figures can be included as multiple sheets in one excel file that is saved using exactly the following convention: S1_Data.xlsx (using an underscore).

-Deposition in a publicly available repository. Please also provide the accession code or a reviewer link so that we may view your data before publication. 

Figure 2E, 3C, S1B-D, S2B, S5

(C) Thank you for depositing the structural and mass spectrometry data in the EMDB (EMD-17391, EMD-17468 and EMD-17467) , EMPIAR (EMPIAR-11557) and PRIDE (PXD042571) data repositories respectively. However, I note that the data is currently on hold. We ask that you please make this data publicly available before publication.

(D) Please also ensure that each of the relevant figure legends in your manuscript include information on *WHERE THE UNDERLYING DATA CAN BE FOUND*, and ensure your supplemental data file/s has a legend.

(E) We require the original, uncropped and minimally adjusted images supporting all blot and gel results reported in the following Figures:

Figure 4C

We will require these files before a manuscript can be accepted so please prepare and upload them now. Please carefully read our guidelines for how to prepare and upload this data: https://journals.plos.org/plosbiology/s/figures#loc-blot-and-gel-reporting-requirements

(F) Please ensure that your Data Statement in the submission system accurately describes where your data can be found and is in final format, as it will be published as written there. 

(G) Please note that per journal policy, the specific species studied (Vairimorpha necatrix) should be clearly stated in the abstract of your manuscript. 

Given the upcoming holiday period, we expect to receive your revised manuscript within 1 month.

*Published Peer Review History*

*Press*

Sincerely,

Richard

Richard Hodge, PhD

rhodge@plos.org,

Reviewer remarks:

Reviewer #1 (Damian Ekiert, signs review): While the authors have mostly responded adequately to my comments, I still object to equating the apparently proteinaceous layer that they observe by cryo-ET with the polar tube proteins (PTPs). This was Major Comment 7 from Reviewer 1, and also Major Comment 1a from Reviewer 2, so clearly this was a conceptual leap that two reviewers objected strongly to. As I said in my original review, it is a reasonable hypothesis that can be discussed, but it is an unsupported and speculative claim. However, while several PTPs have been shown to localize along the length of the PT by IF, the PTP is only ~100 nm is diameter and necessarily diffraction limited. Light microscopy can't readily distinguish what is inside versus outside. Previous EM has suggested an outer protein layer, but what it is made of one can only guess. Additionally, the localization of PTPs along the PT doesn't necessarily mean that they are structural components, or even major components. For example, the layer the authors observed by cryoET may be formed predominantly by a protein that remains to be discovered. The changes to the manuscript don't adequately resolve this problem. For example, lns 111-113 of the revised manuscript state, "...the tube wall, which is composed of a lipid bilayer (pink arrows, Fig. 1 & Supplementary Fig. 2), flanked by an outer layer of polar tube proteins (light blue and magenta arrows, Fig. 1 & Supplementary Fig. 2)." This assignment of the outer layer as consisting of PTPs is not supported by the data and should be removed. Calling it the "Polar tube layer" or "PT layer" also seems to suggest equivalence. I suggest sticking closer to the data and what is known, and referring to it as the "outer layer", "protein layer", or something similar to avoid misleading readers.

Reviewer #2: We thank the authors for their extensive and thoughtful responses, and we found the manuscript to be substantially improved. We feel it is a very nice and timely contribution to the field. We had a few outstanding issues that we would like to point out.

We do disagree with assigning the "PTP layer". This is not supported by data. We feel it is fine to speculate this in the discussion, with a model, which would be very reasonable, but it is confusing for the field when strong conclusions are made which are not adequately supported by data, and this can lead to incorrect assumptions being propagated indefinitely, which is damaging to the field. 

We do feel it is quite confusing to refer to "Empty" tubes that are not empty.

In the methods, we request the authors to please clarify what CNN architecture was used and where it comes from

Just to clarify, our comment regarding whether the lipid bilayer goes all the way around was not referring to the top/bottom anisotropy, but rather to the holes in the side. We were curious as to whether the authors believe there are holes or not, and to please clarify their interpretation for the reader.

Reviewer #3: Through their revisions, the authors were able to significantly improve their manuscript and I have no reservations against its publication in PLOS Biology.

I would like to thank the authors for addressing all of my comments so thoroughly and congratulate them to this fine paper.

---

## [Editor Report · Decision Letter 2]

29 Jan 2024

Dear Dr Barandun,

Thank you for the submission of your revised Short Report entitled "Ultrastructural insights into the microsporidian infection apparatus reveal the kinetics and morphological transitions of polar tube and cargo during host cell invasion." for publication in PLOS Biology. On behalf of my colleagues and the Academic Editor, Nicholas Talbot, I am delighted to let you know that we can in principle accept your manuscript for publication, provided you address any remaining formatting and reporting issues. These will be detailed in an email you should receive within 2-3 business days from our colleagues in the journal operations team; no action is required from you until then. Please note that we will not be able to formally accept your manuscript and schedule it for publication until you have completed any requested changes.

PRESS

Sincerely, 

Ines

--

Ines Alvarez-Garcia, PhD

Senior Editor

PLOS Biology

on behalf of

Richard Hodge, PhD, 

Senior Editor

PLOS Biology

rhodge@plos.org